# Two-Step Contrast Source Learning Method for Electromagnetic Inverse Scattering Problems

**DOI:** 10.3390/s24185997

**Published:** 2024-09-16

**Authors:** Anran Si, Miao Wang, Fuping Fang, Dahai Dai

**Affiliations:** College of Electronic Science and Technology, National University of Defense Technology, Changsha 410073, China; sianran15@nudt.edu.cn (A.S.); wangmiao20@nudt.edu.cn (M.W.); capkoven@mail.ustc.edu.cn (F.F.)

**Keywords:** electromagnetic inverse scattering problems, convolutional neural network, contrast source, dielectric scatterers, image reconstruction

## Abstract

This article is devoted to solving full-wave electromagnetic inverse scattering problems (EM-ISPs), which determine the geometrical and physical properties of scatterers from the knowledge of scattered fields. Due to the intrinsic ill-posedness and nonlinearity of EM-ISPs, traditional non-iterative and iterative methods struggle to meet the requirements of high accuracy and real-time reconstruction. To overcome these issues, we propose a two-step contrast source learning approach, cascading convolutional neural networks (CNNs) into the inversion framework, to tackle 2D full-wave EM-ISPs. In the first step, a contrast source network based on the CNNs architecture takes the determined part of the contrast source as input and then outputs an estimate of the total contrast source. Then, the recovered total contrast source is directly converted into the initial contrast. In the second step, the rough initial contrast obtained beforehand is input into the U-Net for refinement. Consequently, the EM-ISPs can be quickly solved with much higher accuracy, even for high-contrast objects, almost achieving real-time imaging. Numerical examples have demonstrated that the proposed two-step contrast source learning approach is able to improve accuracy and robustness even for high-contrast scatterers. The proposed approach offers a promising avenue for advancing EM-ISPs by integrating strengths from both traditional and deep learning-based approaches, to achieve real-time quantitative microwave imaging for high-contrast objects.

## 1. Introduction

Full-wave electromagnetic inverse scattering problems (EM-ISPs) determine both qualitative and quantitative information about unknown targets within a designated domain of interest (DoI). This non-invasive technique utilizes the scattered electromagnetic field data collected from an illuminating source to infer the properties of objects in the DoI. A significant advantage of employing EM-ISPs lies in their nondestructive nature, which contrasts sharply with traditional methods requiring physical intervention or invasive procedures. Instead, the method allows for the detection of internal heterogeneities by simply observing the scattered fields external to the medium. This attribute makes EM-ISPs highly suitable for applications where the integrity and preservation of the sample or object are paramount, such as remote sensing [1,2], through-the-wall imaging [3], and so on.

Due to the inherent ill-conditioning and nonlinearity of EM-ISPs, the reconstruction methods can be broadly categorized into two principal approaches: regularization iterative optimization methods and non-iterative methods. The conventional iterative approach involves formulating an objective function that encapsulates the problem’s constraints and goals. Subsequently, optimization techniques are employed to iteratively minimize this objective function, quantifying the discrepancy between computed and observed values. By iteratively refining this process, the objective is to achieve convergence and solve EM-ISPs, thereby reconstructing the properties of unknown scatterers. Methods such as the Born iteration method (BIM) [4], the contrast source inversion method (CSI) [5,6,7], and the subspace-based optimization method (SOM) [8,9] commonly encounter several formidable challenges when tasked with reconstructing targets featuring a high dielectric constant. These challenges include sensitivity to the initial selection of values, sluggish convergence rates, elevated computational demands, and susceptibility to becoming trapped in locally optimal solutions. Particularly problematic is their tendency to converge towards false local minima during the inversion process, which becomes notably pronounced when attempting to recover objects characterized by high permittivity. These issues collectively impede their suitability for applications necessitating real-time reconstruction capabilities. Stochastic optimization algorithms, such as genetic algorithm (GA) [10] and particle swarm optimization (PSO) [11], offer advantages in exploring global optimal solutions, due to their inherent uncertainty during iteration. However, these methods are associated with high temporal and spatial complexity. Non-iterative methods simplify the EM-ISPs by ignoring multiple scattering effects and approximating the target as an isolated scattering point, thereby linearizing the problem. Such methods, including Born approximation (BA), Rytov approximation (RA), and back-propagation (BP) [12], are primarily applicable in scenarios characterized by weak scattering (low-contrast targets). In an article by Wang et al. [13], a diffraction tomography (DT) algorithm was introduced for solving 3D EM-ISPs using a sparse planar array and polarization diversity. Nevertheless, non-iterative approaches often fail to achieve successful reconstruction, particularly when dealing with high contrast and large scatterers.

The constraints of conventional methods in computational electromagnetics, such as non-learnable parameters, significant computational overhead, and extensive manual intervention, have driven the rapid adoption of deep learning approaches in this domain [14,15,16,17]. This trend aims to strike a balance between computational efficiency and reconstruction accuracy [18,19,20]. Deep neural networks (DNNs) have the ability to automatically detect features from data; there have been numerous successful applications, including image classification [21] and segmentation [22]. In recent years, learning methods have been a powerful framework enabling unprecedented time and accuracy performance for solving complex EM-ISPs [23,24,25]. Recent developments have increasingly utilized CNNs, with notable success. CNNs employ key operations, such as convolution, addition, ReLU activation, up-sampling, down-sampling, and local maximum filtering. These abilities enable CNNs to learn complex relationships between inputs and outputs [26], and they can also be used to assist iterative optimization methods [27]. In addition to directly mapping scattered fields to scatterer contrasts, using the direct inversion scheme (DIS) [28], another approach involves preprocessing scattering data with iterative or non-iterative inversion algorithms before network training [29,30,31,32]. The dominant current scheme (DCS) [28] first obtains the initial contrast of multiple incident fields from the dominant current and then obtains the contrast through CNN training. This method leverages numerical algorithms to extract prior physical information, enhancing the overall generality and effectiveness of the learning algorithms. Yash Sanghvi et al. introduced the contrast source net (CS-Net) [33], which uses CNNs to derive the contrast source (CS) from scattered field data, followed by traditional iterative algorithms to obtain contrast estimates. These methods typically employ single-network structures. Yao [34] proposed a two-step network structure where the first-step network directly extracts preliminary contrast from scattered field data, refined further in the second-step network. While straightforward and practical, these approaches are purely data-driven and may suffer from limited generalization. In recent years, methods based on deep learning to solve EM-ISPs have flourished [24,32,35,36,37].

Based on prior research and the expressive power of DNNs, we propose a two-step CS learning method for addressing complex EM-ISPs, in order to improve the quality and efficiency of inversion imaging. Specifically, similar to the SOM algorithm, which incorporates subspace decomposition techniques, the CS-Net is employed to learn unknown signal subspaces and recover the complete CS. This recovered CS is then converted into a contrast image, which serves as the input for a second-step U-Net network for refined imaging. Compared with existing conventional methods and two-step DL-based methods [34,38,39,40], the proposed method offers four advantages:(1)The proposed method enables CNNs to manage the entire imaging process without iterative procedures, thereby achieving near-real-time imaging.(2)Despite the inherent non-linearity between the scattered field and object permittivity, the introduction of CSs as intermediate variables in inversion techniques effectively mitigates this issue in EM-ISPs.(3)In the initial phase, integrating physical principles into the CS-Net training enhances noise resilience, incorporates prior physical knowledge, and expands the applicability of the learning algorithms.(4)In the first step, the initial imaging breakthrough allows for only rough imaging of weak scatterers while providing initial imaging of target scatterers with high contrast.

The remainder of this paper is organized as follows: In Section 2, the formulation of the EM-ISPs is introduced. Section 3 outlines the CNN framework and describes the two-step learning method implementation details, involving how to estimate the signal-space components of the CS. Synthetic data are then tested in Section 4 for performance verification. Finally, we summarize this paper and conclude with directions for future extension, in Section 5.

## 2. Problem Formulation

For clarity and ease of explanation, this paper investigates two-dimensional (2D) transverse magnetic (TM, i.e., $E_{z}$ polarization) EM-ISPs. The typical geometric model structure of the 2D-medium free-space electromagnetic inversion imaging system is shown in Figure 1, where D has a free-space background with the permittivity $\epsilon_{0}$ and the scatterer has the relative permittivity $\epsilon_{r}$. Considering the incident case of 2D TM waves, a transmitter generates time-harmonic electromagnetic waves (with a time factor $\left.e^{-i\omega t}\right)$ from different positions around the object, and it irradiates isotropic non-magnetic medium scatterers at a single frequency. We measure the scattered field at different positions on a circular orbit centered around the center of the target area, with receivers distributed equidistant outside the target area. The entire electromagnetic scattering process is represented by two basic electric field integral equations [12] in the observation domain *S* and the imaging domain *D*. For convenience, we represent $\left(\epsilon_{r}\left(r\right)-1\right)$ as the contrast function χ(r) of the scatterer, and $r=\left(x,y\right)$ denotes the source point. We define a CS variable (sometimes referred to as the induced current) as the product of the contrast and the internal field at any point in *D*.

For the sake of our numerical experiments, we solve the discretized version of the Lippmann–Schwinger equation by partitioning DoI into an M×M(M=32) square grid, using the method of moments (MOM) [41]. The discrete form is as follows:(1)$$\bar{J}=\bar{\bar{\chi }}\cdot {\bar{E}}^{\mathrm{tot}\;}$$
(2)$${\bar{E}}^{\mathrm{tot}\;}={\bar{E}}^{\mathrm{inc}\;}+{\bar{\bar{G}}}_{D}\cdot \bar{J}$$
(3)$${\bar{E}}^{\mathrm{sca}\;}={\bar{\bar{G}}}_{S}\cdot \bar{J}$$

Among these, ${\bar{E}}^{\mathrm{tot}\;}$, ${\bar{E}}^{\mathrm{sca}\;}$, and ${\bar{E}}^{\mathrm{inc}\;}$ represent the total, scattered, and incident field, respectively. By subtracting the incident field ${\bar{E}}^{\mathrm{inc}\;}$ from the total field ${\bar{E}}^{\mathrm{tot}\;}$, the scattered field ${\bar{E}}^{\mathrm{sca}\;}$ can be calculated. The scattered field ${\bar{E}}^{\mathrm{tot}\;}$ is measured with $N_{r}$ receivers per illumination, with a total of $N_{t}$ illuminations in a single experiment. $\bar{J}$ is the CS function. The operators $G_{S}\left(\cdot \right)$ and $G_{D}\left(\cdot \right)$ are the mapping from the imaging domain D to the measurement domain *S* and the mapping from the imaging domain *D* to *D*, namely, the Green’s function from the imaging domain D to the receiving antenna *S* and the Green’s function within the imaging domain *D*.

The above three Equations (Equation 1)–(Equation 3) are the basic equations for solving the EM-ISPs. If Ψ is expressed as an operator for solving the corresponding forward problem, the nonlinear relation between the input and output is represented as follows:(4)$${\bar{E}}^{sca}=\Psi \left(\bar{\bar{\chi }}\right)$$

The EM-ISPs estimate $\bar{\bar{\chi }}$, given noisy measurements ${\bar{E}}^{\mathrm{sca}}+\eta$, where η denotes noise, for various illuminations, and, as is typically assumed, ${\bar{E}}^{\mathrm{inc}\;}$ is taken to be known. It can be seen from the above equations that solving the EM-ISPs involves inverting a non-linear and ill-conditioned system of equations between the scattered field and the object contrast. Due to the involvement of many variables in the formula, we have created Table 1 for easy comparison and understanding.

## 3. Theory and Methodology

The equations between the scattered field and the contrast of the object are non-linear and ill-conditioned. Consequently, when solving the EM-ISPs, the system may have infinite solutions, making it challenging to choose a meaningful solution. This issue becomes especially pronounced under conditions involving high-contrast objects or scenes with high frequencies. This method is structured into two sequential steps, depicted in Figure 2. The details of the whole approach are introduced as follows.

### 3.1. Theoretical Background

In EM-ISPs, the full CS is not known, and it cannot be reconstructed directly from the data equation. Therefore, scholars have proposed SOM algorithms that incorporate subspace techniques based on the CSI method. In the SOM, the singular value of the $G_{S}$ operator divides the CS into mutually orthogonal signal and noise subspace components, i.e., ($\bar{J}={\bar{J}}^{s}+{\bar{J}}^{n}$). We perform singular value decomposition (SVD) to obtain ${\bar{\bar{G}}}_{S}\cdot {\bar{\nu }}_{n}=\sigma_{n}{\bar{u}}_{n}$, where ${\bar{\mu }}_{n},{\bar{v}}_{n},\sigma_{n}$ represent the left and right singular vectors and singular values of ${\bar{\bar{G}}}_{S}$, respectively, and $\sigma_{1}\geq \sigma_{2}\cdots \geq \sigma_{M^{2}}\geq 0$. By considering the orthogonality of singular vectors and their relationships,
(5)$${\bar{J}}^{s}=\sum_{j=1}^{L}\frac{{\bar{\mu }}_{j}^{H}\cdot {\bar{E}}^{sca}}{\sigma_{j}}{\bar{v}}_{j}$$
the complete CS can be represented as
(6)$$\bar{J}={\bar{J}}^{s}+V_{N}\alpha$$
where the signal subspace is spanned by the first L right singular vectors and the noise subspace is spanned by the remaining. We need to determine the number of singular values L to be used to define the signal subspace as per Equation (Equation 5). Defining a basis for the latter subspace as $V_{N}=\left[v_{L+1}\ldots v_{M^{2}-L}\right]$, α is the coefficient of the basis vector for the noise subspace, which is as yet unknown. Based on the singular value size corresponding to the signal-to-noise ratio (SNR), we take the right singular vectors corresponding to the first L. Larger singular values form a matrix ${\bar{\bar{V}}}^{+}$, and the remaining M−L right singular vectors can form a matrix ${\bar{\bar{V}}}^{-}$. L is chosen such as to avoid fitting the noise. In the SOM, the unknown parameters α and $\bar{\bar{\chi }}$ are updated alternatively.

Neural networks are highly effective at parallel computing, making them well-suited to addressing large-scale EM-ISPs. CNNs excel in analyzing boundary changes in input data using convolutional modules, and they are particularly adept at handling matrix-structured data. Their robust nonlinear fitting capabilities allow for the establishment of complex one-to-one mappings between input and output images, facilitating accurate regression on image-like data. Consequently, the EM-ISPs can be modeled as an input–output mapping relationship. By training a neural network model with a series of data samples, we can tackle nonlinear issues while benefiting from the model’s strong robustness against noise and uncertainties in the input data.

### 3.2. Initial Guess (Step 1)

To begin, we employ the CS-Net framework to learn the noise spatial component of the radiation operator. The CS-Net is trained specifically for this purpose. Subsequently, the coarse contrast is derived through CS-based restoration. The architecture details of the CS-Net, employed for the estimation of the noise subspace components, are illustrated in Figure 3:

Similar to the method in the SOM, we first perform SVD based on Green’s function operator $G_{S}$. Therefore, as an initial guess of the CS, we use the signal subspace component ${\bar{J}}^{s}$, which contains the most important information but lacks some information contained in ${\bar{J}}^{n}$. To compensate the missing information ${\bar{J}}^{n}$, the signal subspace component ${\bar{J}}^{s}$ is used as input to the CS-Net in the first step. Since the ${\bar{J}}^{s}$ is complex-valued, we separate its real and imaginary parts as two channels. Thus, the input of the CS-Net is a $2*N_{i}$-channel real-valued concatenation of ${\bar{J}}^{s}$. The CS-Net aims to reconstruct the full CS $\bar{J}$ from the input ${\bar{J}}^{s}$ for each incidence. After the CS-Net predicts all the CS $\bar{J}$ for each incidence, the predicted contrast can be calculated by the basic Equations (Equation 1)–(Equation 3). Each emission scenario generates a corresponding CS image, which is utilized to train the CS-Net for recovering the full CS. The network is trained using the average mean squared loss between the estimated and the true CS, and the network parameters are optimized using the Adam optimizer with learning rate $10^{-4}$. Afterwards, the restored CS is used for SOM imaging, to compare with our proposed two-step CS learning method. The optimization process terminates when either the relative change in cost function drops below $10^{-4}$ or after 2000 iterations, whichever comes first.

### 3.3. Fine Imaging (Step 2)

In the second step, the U-Net [42] takes as input the coarse contrast image recovered by the CS-Net and generates an improved estimate of the contrast image. The U-Net model for the second step is benchmarked in MATLAB 2019a, using the Deep Learning Toolbox. An adaptive moment estimation optimizer is employed, to minimize the half-mean-squared-error loss function, while a dropout regularization rate of 0.2 is applied, to reduce overfitting. Due to the down-sampling operations and batch normalization (BN) structure in the U-Net CNN architecture, it is particularly well-suited for addressing EM-ISPs [43]. The chosen loss function is the mean square error (MSE), calculated between the true labels and the predicted pixel responses. The architecture details of the U-Net used for fine imaging are described in detail in Figure 4. Therefore, the initially retrieved result is further enhanced by the U-Net, to achieve improved reconstruction. However, the potential prior information acquired by the CS-Net and the U-Net throughout the two-step process has yet to be fully understood and explored.

### 3.4. Image Evaluation

To compare different schemes, we utilize three key quantitative metrics: the structural similarity index measure (SSIM) [44], the peak signal-to-noise ratio (PSNR), and the equivalent number of looks (ENL). These metrics serve as indicators to assess the fidelity of reconstructed contrast images. Among them, higher SSIM and PSNR values indicate greater similarity and higher quality between the true profile image and its reconstructed counterpart. Conversely, a lower ENL indicates improved clarity and reduced noise in the image.

### 3.5. Computational Complexity

Before the network training begins, performing a thin SVD on the matrix ${\bar{\bar{G}}}_{S}$ has the computational complexity of $O\left(N_{r}^{2}M^{2}\right)$. In the first step, the CS-Net is primarily responsible for reconstructing the complete CS, with a relatively lightweight network architecture designed to handle the determined part of the CS. The computational complexity of this step is mainly due to the convolution operations, with a time complexity of approximately $O\left(NM^{2}K\right)$, where N is the number of samples, $M^{2}$ is the size of the input feature maps, and K is the number of convolutional kernels. In the second step, the U-Net is used to further optimize the contrast images. Although the U-Net has relatively higher computational complexity, the use of skip connections and multi-scale feature-extraction mechanisms effectively reduces the computational burden. The complexity of the U-Net is primarily influenced by convolution and pooling operations, with time complexity expressed as $O\left(pq^{2}r\right)$, where p is the number of samples, $q^{2}$ is the size of the input feature maps, and r is the number of convolutional kernels. Such a computation can be accelerated by using GPU. In contrast, the traditional SOM has computational complexity $O\left(IN_{i}M^{2}\log M\right)$, where *I* is the total number of iterations, $N_{i}$ is the number of illuminations, and $M^{2}$ is the number of pixels in the imaging domain. Each gradient and step calculation for a single view requires $O\left(M^{2}\log M\right)$ rather than $O\left(M^{4}\right)$, due to the use of fast Fourier transform (FFT) operations for all matrix-vector computations within the matrix ${\bar{\bar{G}}}_{D}$.

## 4. Numerical Example

To assess the effectiveness and accuracy of the proposed two-step CS learning approach, experiments were conducted using the Modified National Institute of Standards and Technology (MNIST) dataset [45]. The MNIST dataset consists of images of handwritten digits and is well-standardized with a broad range of applications. Although it does not directly represent EM-ISPs, using this dataset allows for testing the algorithm’s fundamental image processing and generalization capabilities. Specifically, the testing samples were digit images extracted from the MNIST test dataset that had not been encountered by the CS-Net and U-Net during the training phase. This section presents our results, using synthetic data to assess the performance of the proposed two-steps approach in reconstructing contrast from the scattered field. In all the tests, the reconstructions were performed at a fixed frequency, without employing frequency-hopping techniques. Moreover, as the comparison, the two-step enhanced deep learning approach [34] was also employed, to reconstruct the contrast of the testing samples. All the numerical experiments in this paper were performed on an Intel(R) Xeon(R) processor running at 2.10 GHz with 128 GB RAM.

### 4.1. Configuration of the Scattering System

In our numerical experiments, we utilized the MNIST database to generate a scattered field, using a forward solver [41]. The scattered field was intentionally corrupted with additive Gaussian noise during each reconstruction. Under our experimental conditions, synthetic noise with a signal-to-noise ratio (SNR) of 25 dB was applied to the scattered field. Consequently, the optimal number of singular values, denoted as L=19, was determined, based on Morozov’s discrepancy principle [46]. This optimal choice varied with the SNR levels observed in the measurements. Based on MNIST, the contrast of the number-shaped objects was set between 1.0 and 7.0 with a free space background. Throughout, the incident field frequency was 400MHz (i.e., wavelength λ=0.75m). Measurements were taken along a circular path with a radius R=4m, centered precisely on the DoI. The measurement setup included $N_{r}=32$ receivers and $N_{t}=16$ transceivers uniformly distributed in an equi-angular manner across the measurement domain.

### 4.2. Test Using MNIST Database with SNR = 25 dB

In the first example, the scattered field was corrupted by additive Gaussian noise, such that the SNR was 25 dB. The reconstruction results of six randomly selected examples, i.e., labeled Test(1) through Test(6), are displayed in Figure 5. Quantitatively, the reconstructed results of the different methods were calculated, and the corresponding image quality metrics ENL, PSNR, and SSIM are listed in Table 2. Figure 5 shows the ground truth images of the testing samples alongside the reconstructed images, using different methods. These included the SOM with the CS recovered by the CS-Net, the first and second steps of the two-step enhanced deep learning approach by Yao et al. [34], and the first and second steps of our proposed two-step CS learning method. Clearly, the final outputs of the proposed method were a much better reconstruction of the ground truth. Table 2 shows that incorporating prior labeled data enhanced the reconstruction quality, surpassing both the SOM and the two-step enhanced deep learning method in accuracy and efficiency. The contrast image converted from the CS restored by the CS-Net shows that although the contrast value of the target scatterer and the edge reconstruction of the contrast image were still not accurate enough, the approximate position and contour of the target scatterer could be vaguely displayed from it. It should also be noted that the metrics of the proposed method in the second row (Test 2) slightly outperformed those of the SOM. However, challenges include potential local minima issues when using the CS restored by the CS-Net for SOM imaging. Figure 6 shows the statistical analysis of the testing results, confirming that our proposed two-step CS learning method significantly improves performance in solving EM-ISPs.

### 4.3. Test Using MNIST Database with SNR = 15 dB

To verify the robustness of the method, in the second example, we also tested the data target with an SNR of 15 dB, i.e., the scattered field was corrupted with additive Gaussian noise, such that the SNR was 15 dB. The results depicting the reconstruction of Test(1)–Test(6) are presented in Figure 7, while the corresponding quality metrics, SSIM, PSNR, and ENL are detailed in the accompanying Table 3, and the statistical analysis of the testing results is presented in Figure 8. Based on the imaging results and a comprehensive analysis of the three evaluation indicators, it is evident that for scatterers with a contrast exceeding 2.0, the proposed method demonstrated superior performance compared to the SOM imaging, in terms of both reconstructed contours and image clarity. Compared to the two-step enhanced deep learning approach [34], our proposed CS learning method exhibited stronger robustness. It should also be noted that the contrast imaging result of our proposed method in the first row (Test 1), with a contrast value of 2.0, was slightly better than that of the SOM. However, while the SOM method requires thousands of iterations and an imaging time of four to five minutes, our method can achieve near-real-time imaging.

### 4.4. Test Using Austria Profile with SNR = 25 dB

Next, the proposed scheme was evaluated, using the Austria profile with a contrast value of 2.0, which consisted of one central ring and two disks. The results depicting the reconstruction of the Austria profile are presented in Figure 9. The evaluation index values for image reconstruction are presented in Table 4. From the imaging results, the use of the CS recovered by the CS-NET for SOM imaging of low-contrast Austria targets shows promising outcomes. However, the SOM required thousands of iterations, resulting in prolonged imaging times exceeding five minutes. In contrast, the proposed two-step CS method achieved near-real-time imaging capabilities.

## 5. Conclusions

In this study, we propose a novel two-step CS learning approach to solving EM-ISPs. Our method employs a two-step process for CS and contrast image reconstruction, validated through simulation data that demonstrate effective reconstruction capabilities, feasibility, and efficiency. Initially, the CS-Net integrates physical insights to restore the complete CS and converts the predicted complete CS output into a contrast image, which is then used as input for the U-Net. The U-Net refines these initial results, leveraging the previously obtained rough contrasts, to progressively improve the image quality from coarse to fine. This refinement process progresses from rough intermediate images towards fine images. Utilizing prior information in the CS-Net noticeably improves reconstruction outcomes. Essentially, the U-Net enhances the initial reconstruction, yielding clearer and more accurate images. As a result, our approach achieves significantly improved accuracy in solving EM-ISPs, especially for high-contrast scatters (up to a contrast of 7.0). The proposed approach offers a promising avenue for advancing EM-ISPs by integrating strengths from both traditional and deep learning-based approaches, to achieve real-time quantitative microwave imaging. The parameterization of EM-ISPs using DNNs facilitates GPU-friendly processing, due to high parallelization. Interpretability remains a concern in CNN steps, affecting inversion accuracy, which relies on the proximity of input image quality to the reference. Future work could focus on enhancing the CS-Net model structure and refining the training parameters to improve noise subspace estimation and reduce the occurrence of local minima. Additionally, future research should focus on optimizing computational efficiency through improved model initialization and GPU parallelization. While the MNIST database was used to test the effectiveness of this method in terms of simplicity and generality, its direct applicability to EM-ISPs is limited, as it does not represent the real physical properties of scattering objects. Future work should involve testing with data that more accurately reflects actual EM-ISPs. Moreover, although the results of the proposed method are promising, further research is needed, particularly in enhancing its limited aperture-inversion capability and adapting it to 3D scenes.

## Figures and Tables

**Figure 1 sensors-24-05997-f001:**
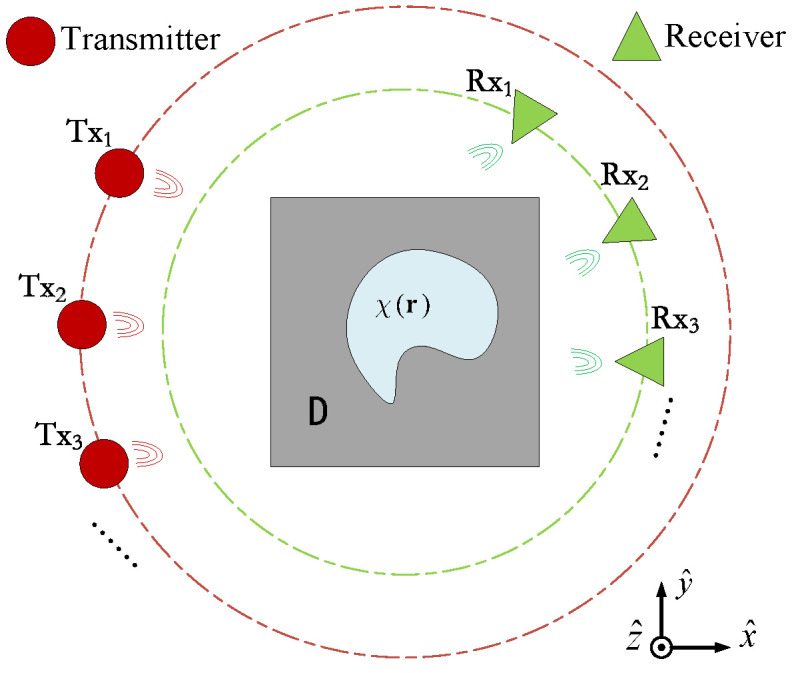
Geometry of EM-ISPs in 2D case with TM illuminations.

**Figure 2 sensors-24-05997-f002:**
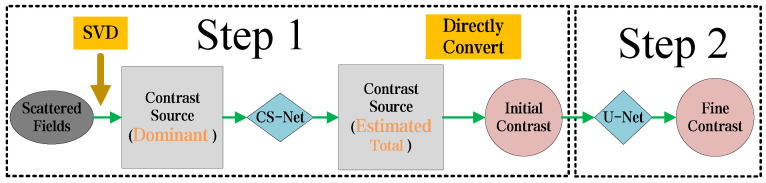
Overall flowchart of the proposed two-step CS learning method for EM-ISPs. In Step 1, the Green’s function operator is subjected to SVD based on scattered field data. The dominant CS obtained from SVD is used as the input for the CS-Net, which generates an initial estimate of the total CS. This estimated CS is then directly converted into a contrast image. In Step 2, the coarse contrast image is refined, using the U-Net to obtain the final solution.

**Figure 3 sensors-24-05997-f003:**
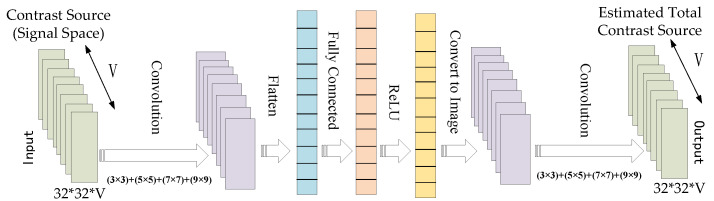
Architecture for CS-Net in Step 1. The input of the CS-Net is the signal subspace of the CS, which is represented as 32 × 32 image with V = 32 channels. The output of the CS-Net is an estimate of the true CS. The first layer performs convolution with different filter sizes, i.e., (3 × 3), (5 × 5), (7 × 7), and (9 × 9), each with eight channels, and stacks the filter activations, to form a 32 × 32 image with 32 channels. The image is then vectorized and passed through three fully connected layers, each with a ReLU activation. The output vector is reshaped to a 16-channel 32 × 32 image again, and a last convolution layer is used to generate the estimate of the true CS.

**Figure 4 sensors-24-05997-f004:**
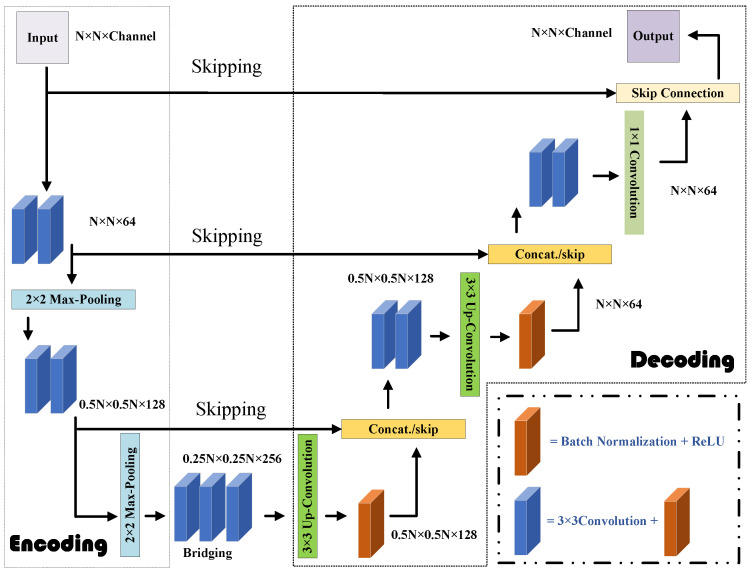
Architecture for the U-Net in Step 2. The encoding part is equipped with the repeated application of 3 × 3 convolution, BN, and a rectified linear unit (ReLU) and 2 × 2 max-pooling operation. Meanwhile, the decoding part is armed with the repeated application of 3 × 3 up-convolution, BN, ReLU, and a concatenation operation with skip connection.

**Figure 5 sensors-24-05997-f005:**
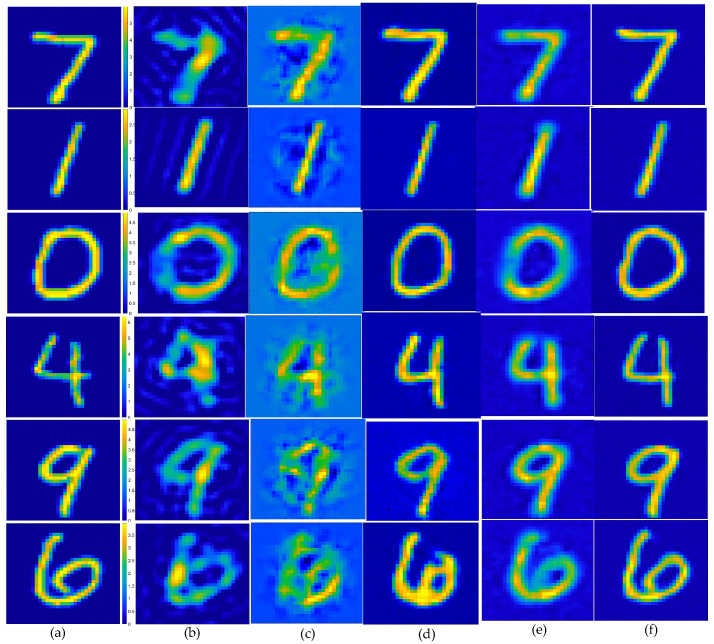
Estimated contrast image on MNIST test images, SNR = 25 dB: (**a**) ground truth; (**b**) reconstruction results of object from the SOM, using the CS recovered by the CS-Net; (**c**) reconstructed rough contrast image from the first step of the two-step enhanced deep learning approach [34]; (**d**) reconstructed final contrast image from the second step of the two-step enhanced deep learning approach [34]; (**e**) direct conversion of the CS recovered by the CS-Net into contrast imaging results; (**f**) outperforming that of the U-Net.

**Figure 6 sensors-24-05997-f006:**
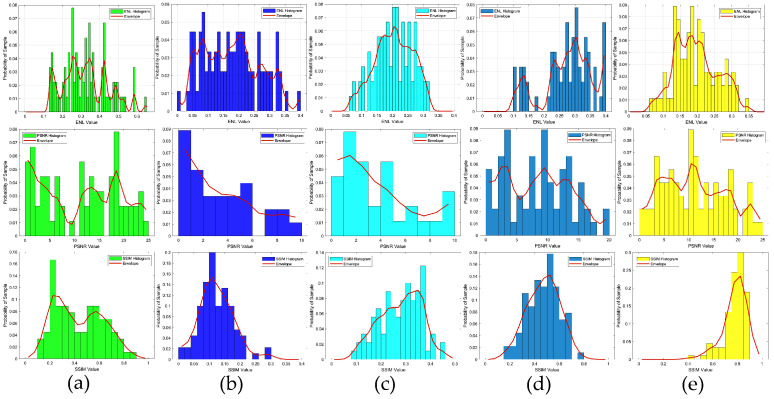
ENL, PSNR, and SSIM statistical histograms of the reconstructed contrast image quality, SNR = 25 dB: (**a**) results obtained from the SOM using the CS recovered by the CS-Net; (**b**) results obtained from the first step of the two-step enhanced deep learning approach [34]; (**c**) results obtained from the second step of the two-step enhanced deep learning approach [34]; (**d**) results obtained from the first step of the proposed two-step CS learning method; (**e**) results obtained from the second step of the proposed two-step CS learning method.

**Figure 7 sensors-24-05997-f007:**
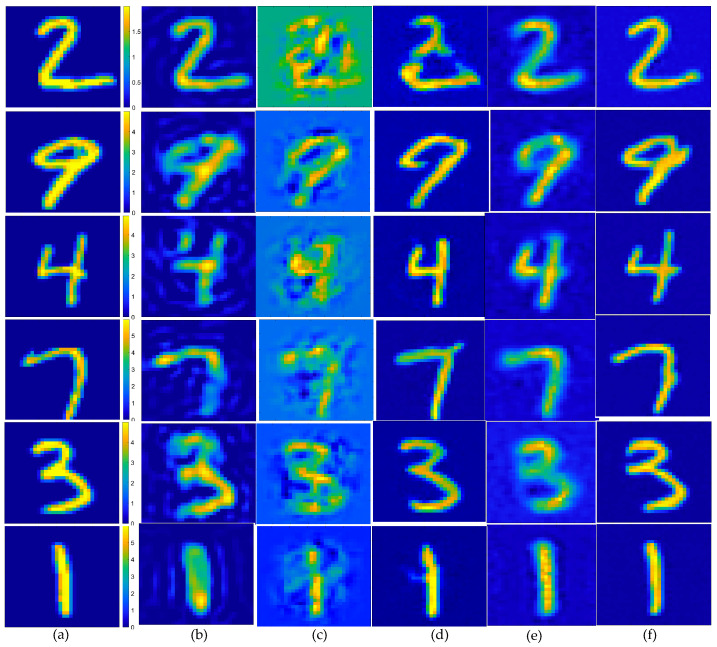
Estimated contrast image on MNIST test images, SNR = 15 dB: (**a**) ground truth; (**b**) reconstruction results of object from the SOM, using the CS recovered by the CS-Net; (**c**) reconstructed rough contrast image from the first step of the two-step enhanced deep learning approach [34]; (**d**) reconstructed final contrast image from the second step of the two-step enhanced deep learning approach [34]; (**e**) direct conversion of the CS recovered by the CS-Net into contrast imaging results; (**f**) outperforming that of the U-Net.

**Figure 8 sensors-24-05997-f008:**
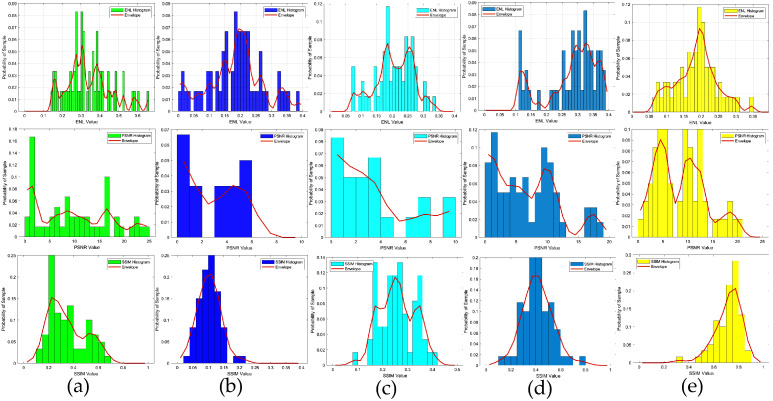
ENL, PSNR and SSIM statistical histograms of the reconstructed contrast image quality, SNR = 15 dB: (**a**) results obtained from the SOM using the CS recovered by the CS-Net; (**b**) results obtained from the first step of the two-step enhanced deep learning approach [34]; (**c**) results obtained from the second step of the two-step enhanced deep learning approach [34]; (**d**) results obtained from the first step of the proposed two-step CS learning method; (**e**) results obtained from the second step of the proposed two-step CS learning method.

**Figure 9 sensors-24-05997-f009:**
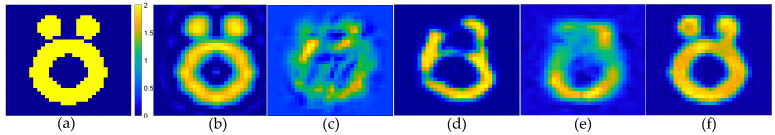
Estimated contrast image on Austria profile, SNR = 25 dB: (**a**) true Austria profile; (**b**) reconstruction results of object from the SOM using the CS recovered by the CS-Net; (**c**) reconstructed rough contrast image from the first step of the two-step enhanced deep learning approach [34]; (**d**) reconstructed final contrast image from the second step of the two-step enhanced deep learning approach [34]; (**e**) direct conversion of the CS recovered by the CS-Net into contrast imaging results; (**f**) outperforming that of the U-Net.

**Table 1 sensors-24-05997-t001:** For the $p\mathrm{th}\;(p=1,2,\ldots ,N_{t})$ illumination, the relevant variables used in the above formula.

Variable Name	Size	Meaning
${\bar{E}}^{\mathrm{tot}\;}$	$\in C^{M^{2}\times 1}$	The total internal fields data on domain S.
${\bar{E}}^{\mathrm{inc}\;}$	$\in C^{M^{2}\times 1}$	The incident fields data in the DoI.
${\bar{E}}^{sca}$	$\in C^{N_{r}\times 1}$	The scattered fields data on domain S.
$\bar{J}$	$\in C^{M^{2}\times 1}$	The CS function in the DoI.
$\bar{\bar{\chi }}$	$\in C^{M^{2}\times 1}$	The diagnonal matrix of the contrast function.
${\bar{\bar{G}}}_{S}$	$\in C^{N_{r}\times M^{2}}$	The radiation operators (the DoI to the domain S).
${\bar{\bar{G}}}_{D}$	$\in C^{M^{2}\times M^{2}}$	The radiation operators (the DoI to the DoI).

**Table 2 sensors-24-05997-t002:** Metrics of different methods, SNR = 25 dB.

	Metric	(b)	(c)	(d)	(e)	(f)
**Test(1): first row digit “7”**	**SSIM**	0.2201	0.1064	0.2836	0.4003	0.7175↑
	**PSNR**	14.1030	11.8024	12.3382	18.2844	24.4498↑
	**ENL**	0.3615	0.1303	0.1754	0.2353	0.1433↓
**Test(2): second row digit “1”**	**SSIM**	0.3532	0.1722	0.3311	0.5839	0.8924↑
	**PSNR**	22.9435	17.6328	14.1315	21.5712	30.3555↑
	**ENL**	0.1390	0.0487	0.0767	0.1070	0.0760↓
**Test(3): third row digit “0”**	**SSIM**	0.2861	0.2054	0.3219	0.5883	0.7931↑
	**PSNR**	19.7904	11.9012	12.2882	22.8354	24.7904↑
	**ENL**	0.5392	0.2267	0.2296	0.3488	0.2241↓
**Test(4): fourth row digit “4”**	**SSIM**	0.2369	0.1621	0.3483	0.4007	0.7147↑
	**PSNR**	17.9987	13.2725	14.0838	14.6851	19.2040↑
	**ENL**	0.3461	0.2015	0.1606	0.2630	0.1520↓
**Test(5): fifth row digit “9”**	**SSIM**	0.6835	0.2170	0.3672	0.6761	0.8810↑
	**PSNR**	23.4289	15.2951	17.0325	19.4508	28.7113↑
	**ENL**	0.2665	0.1842	0.2059	0.2910	0.1988↓
**Test(6): sixth row digit “6”**	**SSIM**	0.2442	0.1957	0.3692	0.3248	0.8093↑
	**PSNR**	19.4901	16.8602	17.4904	18.5999	21.6365↑
	**ENL**	0.4229	0.3294	0.3005	0.4351	0.2793↓

**Table 3 sensors-24-05997-t003:** Metrics of different methods, SNR = 15 dB.

	Metric	(b)	(c)	(d)	(e)	(f)
**Test(1): first row digit “2”**	**SSIM**	0.5584	0.0837	0.1613	0.4605	0.7858↑
	**PSNR**	24.4427	9.1308	10.5866	18.2878	25.2928↑
	**ENL**	0.3148	0.0950	0.2762	0.3246	0.2074↓
**Test(2): second row digit “9”**	**SSIM**	0.2649	0.2029	0.3932	0.3721	0.7834↑
	**PSNR**	17.6125	13.9164	15.1215	19.9382	24.2291↑
	**ENL**	0.4853	0.1522	0.2116	0.3396	0.2251↓
**Test(3): third row digit “4”**	**SSIM**	0.1715	0.0825	0.2969	0.3782	0.6624↑
	**PSNR**	14.6102	13.2556	13.5544	19.4833	25.4906↑
	**ENL**	0.3144	0.2008	0.1457	0.2521	0.1836↓
**Test(4): fourth row digit “7”**	**SSIM**	0.2460	0.0425	0.1987	0.3335	0.7619↑
	**PSNR**	23.0121	10.8452	10.3681	19.3680	27.3919↑
	**ENL**	0.2694	0.1246	0.1618	0.2369	0.1318↓
**Test(5): fifth row digit “3”**	**SSIM**	0.2606	0.1185	0.2410	0.3281	0.6867↑
	**PSNR**	15.3602	12.0076	12.4106	17.9613	21.3712↑
	**ENL**	0.4859	0.2035	0.2214	0.3627	0.2007↓
**Test(6): sixth row digit “1”**	**SSIM**	0.2030	0.0949	0.2745	0.3910	0.7850↑
	**PSNR**	18.8818	7.6736	5.4605	22.9497	24.5452↑
	**ENL**	0.2282	0.2142	0.1063	0.1433	0.0936↓

**Table 4 sensors-24-05997-t004:** Metrics comparing different methods for the Austria profile, SNR = 25 dB.

	Metric	(b)	(c)	(d)	(e)	(f)
**Austria:**	**SSIM**	0.5787	0.2834	0.4001	0.3960	0.8195↑
	**PSNR**	16.0104	15.9636	16.8300	17.4791	19.0318↑
	**ENL**	0.5823	0.4490	0.3048	0.5658	0.4268↓

## Data Availability

The data presented in this study are available on request from the corresponding author.

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
