# Peer review of "Two-Step Contrast Source Learning Method for Electromagnetic Inverse Scattering Problems"

_sensors, 2024, doi:10.3390/s24185997_

Round 1
Reviewer 1 Report
Comments and Suggestions for Authors
Summary:
The article presents a solution for solving electromagnetic inverse scattering problems (EM-ISPs) using a two-step deep learning approach. However, there are some areas that need improvement to enhance the coherence and cohesion of the text, as well as to ensure that the contributions of the research are clearly communicated.
Strengths:
1. The proposed methodology, which integrates convolutional neural networks (CNNs) in a two-step process, is innovative and promises advancements in terms of accuracy and efficiency in image reconstruction for EM-ISPs.
2. The structure of the article is clear and well-organized, making the presented concepts easy to understand.
Areas for Improvement:
1. Clarity and Motivation in the Abstract:
- Comment: The abstract needs to clarify the context and motivation of the research. This will help readers better understand the relevance of the work.
- Suggestion: Rewrite the abstract to include a brief explanation of the importance of improving reconstruction techniques in EM-ISPs.
2. Keyword ‘Two-step Method’:
- Comment: The keyword "Two-step Method" is not specific enough.
- Suggestion: Replace it with a more relevant keyword,.
3. Related Works Comparison Section:
- Comment: The comparison with previous works is not clearly separated.
- Suggestion: Create a separate section to discuss the differences between the proposed work and recent research, including references such as:
ZHANG, Huan Huan et al. Enhanced two-step deep-learning approach for electromagnetic-inverse-scattering problems: Frequency extrapolation and scatterer reconstruction. IEEE Transactions on Antennas and Propagation, v. 71, n. 2, p. 1662-1672, 2022.
YAO, He Ming; NG, Michael; JIANG, Lijun. Deep Learning Electromagnetic Inversion Solver Based on Two-Step Framework for High-Contrast and Heterogeneous Scatterers. IEEE Transactions on Antennas and Propagation, 2024.
SI, Anran et al. Two Steps Electromagnetic Quantitative Inversion Imaging Based on Convolutional Neural Network. In: 2024 5th International Conference on Geology, Mapping and Remote Sensing (ICGMRS). IEEE, 2024. p. 28-32.
4. Methodology Summary in the Introduction:
- Comment: The introduction lacks a concise summary of the methodology.
- Suggestion: Include a paragraph summarizing the methodological approach to provide an overview before the Theory and Methodology section.
5. Theoretical Background Section:
- Comment: Details about the theoretical concepts used in the methodology are missing.
- Suggestion: Add a section called "Theoretical Background" to explain the key concepts, such as convolutional neural networks and electromagnetic inversion methods.
6. Comparison of Results with Recent Methods:
- Comment: The comparison of results could be more detailed.
- Suggestion: Include a detailed comparison of metrics with recent methods and add bar charts to better visualize the differences.
7. Improve Tables 2, 3, and 4:
- Comment: Tables 2, 3, and 4 could be more informative and visually clear.
- Suggestion: Revise the formatting of these tables and consider adding charts to complement the data.
Comments on the Quality of English Language
- Comment: Some passages in the text lack coherence and cohesion.
- Suggestion: Rewrite sentences to improve the flow of the text. Examples include:
a. Original: "Subsequently, these initial results are refined using U-Net, effectively utilizing rough contrasts obtained previously."
Improvement: "These initial results are then refined using U-Net, which effectively utilizes the previously obtained rough contrasts."
b. Original: "It also needs to note that the metrics of the proposed method in the second row (Test 2) are slightly better than those of the SOM."
Improvement: "It should also be noted that the metrics of the proposed method in the second row (Test 2) slightly outperform those of the SOM.
c. Original: "However, the potential prior information acquired by CS-Net and U-Net throughout the two-step process remain to be fully comprehended and explored."
Improvement: "However, the potential prior information acquired by CS-Net and U-Net throughout the two-step process has yet to be fully understood and explored."
d. Original: "Experimental results demonstrate successful reconstruction of high-contrast scatters (up to contrast 7.0). Our method enhances both image quality and computational efficiency, addressing challenging non-linear problems."
Improvement: "The experimental results demonstrate successful reconstruction of high-contrast scatterers (up to a contrast of 7.0). Our method improves both image quality and computational efficiency, successfully addressing challenging nonlinear problems."
e. Original: "For sake of numerical experiments, we solve the discretized version of the Lippmann–Schwinger equation by partitioning DoI into an M×M (M = 32) square grid using the method of moments (MOM)."
Improvement: "For the sake of numerical experiments, we solve the discretized version of the Lippmann–Schwinger equation by partitioning the DoI into an M×M (M = 32) square grid using the method of moments (MoM)."
f. Original: "It is evident that utilizing CS recovered by CS-NET for SOM imaging of low-contrast Austria targets yields promising outcomes."
Improvement: "The use of CS recovered by CS-NET for SOM imaging of low-contrast Austrian targets has shown promising outcomes."
g. Original: "(…) introducing CS as intermediate variables in inversion techniques effectively mitigates this issue in EM-ISPs."
Improvement: "(…) the introduction of CS as intermediate variables in inversion techniques effectively mitigates this issue in EM-ISPs."
h. Original: "The proposed method facilitates CNNs to handle the entire imaging process without iterative procedures, thereby achieving nearly real-time imaging."
Improvement: "The proposed method enables CNNs to manage the entire imaging process without iterative procedures, thereby achieving near real-time imaging."
i. Original: "Higher values of SSIM and PSNR signify greater similarity and higher quality between the true profile image and its reconstructed counterpart."
Improvement: "Higher SSIM and PSNR values indicate greater similarity and higher quality between the true profile image and its reconstructed counterpart."
Author Response
Thank you very much for taking the time to review our manuscript. We appreciate the efforts of the editors and reviewers in providing feedback and are grateful for their insightful comments and valuable suggestions. We have carefully revised the paper based on all the feedback received. We look forward to your further review and guidance.

Reviewer 2 Report
Comments and Suggestions for Authors
In this work, the authors developed a noval two-step contrast source learning approach for electromagnetic inverse scattering problems, Numerical Example show the proposed method enhances reconstruction accuracy and efficiency over traditional methods. This work could be accepted in present form.
The article is well-organized, with a clear line of thought, citing a series of relatively recent relevant literature. The proposed method is technically sound and offers a clear implementation description. No errors were found in the tables and formulas, and the selected numerical examples effectively demonstrate the effectiveness of the method. Compared to traditional methods, this method shows significant improvement in SSIM, PSNR, and ENL metrics.
Author Response
Thank you very much for taking the time to review this manuscript. We look forward to your further review and guidance.
Reviewer 3 Report
Comments and Suggestions for Authors
1. In my opinion, the authors should write the summarize the important results (in numeric) in the Abstract or Conclusions sections.
2. Please give more details the theory or background knowledge or governed equations. (if any)
3. Do the authors compare these results to other methods in each case?
Author Response

(The authors gave the same response as above.)

Reviewer 4 Report
Comments and Suggestions for Authors
This paper proposes propose a two-step contrast source learning approach, cascaded convolutional neural networks (CNNs) into the inversion framework, to tackle the 2-D full-wave EMISPs. The following questions need to be answered.
1.Figures 2-4 need to be described in detail to facilitate a better understanding of the proposed methodology by the authors.
2. How about the computational complexity?
3. Please clarify the differences between the proposed method and existing learning-based networks.
4. The contributions of this paper are not so clear to the reviewer. Please clarify which one is existing and which one is your own?
5. The proposed method suggests adding more details.
6. The experiments should be given more to show the effectiveness of the proposed method.
Comments on the Quality of English Language
The English Language is good.
Author Response

(The authors gave the same response as above.)

Reviewer 5 Report
Comments and Suggestions for Authors
1. When future research is mentioned in the conclusion, some more specific research questions or application scenarios can be proposed based on current trends in the field.
2. The introduction mentions the shortcomings of some traditional methods, such as BIM, CSI, and SOM, but there is no detailed review of the latest research progress. It is recommended that the authors add new developments in electromagnetic inverse scattering problems in recent years, especially the application of deep learning and hybrid methods, to enhance the comprehensiveness and cutting-edge nature of the literature review.
3. In the experimental part, the article uses the MNIST data set for testing, but does not explain why this data set was chosen. The authors are recommended to discuss the characteristics of the MNIST dataset and how it represents practical application scenarios for electromagnetic inverse scattering problems.
4. The article mentioned the cases of SNR=25 dB and 15 dB, but did not discuss the impact of other possible noise levels on model performance. It is recommended to add experiments at different SNR levels, or conduct a rational analysis of the currently selected SNR value.
5. The format and name of the figure 1 picture are not centered.Page 5 has formatting problems, and the first line is not indented.
6. The introduction gives a comprehensive overview of the article in clarity and structure, but it would be useful to explain more clearly how the proposed two-step method is fundamentally different from existing methods. This helps to highlight the novelty and contribution of this paper.
7. The methodology section explains the steps involved well, but not in detail how the parameters of CNN and U-Net [] were chosen. In-depth discussion of the selection and tuning of these parameters can enhance the technical depth of this paper.
8. On experimental results, while promising, this paper can benefit from a more detailed comparison with other state-of-the-art methods, especially in terms of computational efficiency and accuracy. Including a table or graph directly comparing these indicators will strengthen the demonstration of the validity of the proposed method.
Comments on the Quality of English Language
It can be properly polished.
Author Response

(The authors gave the same response as above.)

Round 2
Reviewer 1 Report
Comments and Suggestions for Authors
I appreciate the efforts made to address the previous comments, and we acknowledge the significant improvements implemented. Despite these improvements, there are still some areas that require further attention to ensure the methodological rigor and clarity of the manuscript. Below are the key points that should be addressed:
1. Structure of Section 2:
We recommend not starting Section 2 with a figure. This disrupts the flow of the text. It would be better to include a brief introduction or context before presenting any figure.
2. Computational Complexity:
The section on computational complexity mentions that UNet and CSNet have their own computational demands, but it lacks a clear comparison with traditional methods such as the Subspacebased Optimization Method (SOM). A more detailed comparison would help quantify the efficiency of the proposed method in terms of computational cost.
Additionally, the manuscript claims that the method achieves "near realtime" reconstruction, but it does not provide specific details on execution times or the hardware configuration used. These details are crucial for evaluating the practical applicability of the method, and we suggest that this information be included.
3. Validation Using MNIST Database:
While the use of the MNIST database for testing the method is valid in terms of simplicity and generalization, MNIST is typically used for digit classification tasks. The direct applicability of MNIST to electromagnetic scattering problems is limited, as it does not reflect the real physical properties of a scattering object. We recommend that you mention this limitation explicitly in the conclusions and acknowledge that future work should include testing with data more representative of actual electromagnetic scattering problems.
In conclusion, the article has been significantly improved, but addressing these points would further strengthen the methodological rigor and clarity of the presentation.
Author Response
Thank you for your thorough review and valuable feedback.

Reviewer 4 Report
Comments and Suggestions for Authors
The author have answered my questions.
Comments on the Quality of English Language
The English Language is good.
Author Response
Thank you for your meticulous review.

Reviewer 5 Report
Comments and Suggestions for Authors
The authors have addressed the comments from the reviewer. Acceptance is recommended.
Comments on the Quality of English Language
English is good.
Author Response
Thank you for your meticulous review.
